# TOWARDS ANTISYMMETRIC NEURAL ANSATZ SEPARATION

## ABSTRACT

We study separations between two fundamental models (or *Ansätze*) of anti-symmetric functions, that is, functions $f$ of the form $f(x_{\sigma(1)}, \ldots, x_{\sigma(N)}) = \text{sign}(\sigma) f(x_1, \ldots, x_N)$, where $\sigma$ is any permutation. These arise in the context of quantum chemistry, and are the basic modeling tool for wavefunctions of Fermionic systems. Specifically, we consider two popular antisymmetric Ansätze: the Slater representation, which leverages the alternating structure of determinants, and the Jastrow ansatz, which augments Slater determinants with a product by an arbitrary symmetric function. We construct an antisymmetric function that can be more efficiently expressed in Jastrow form, yet provably cannot be approximated by Slater determinants unless there are exponentially (in $N^2$) many terms. This represents the first explicit quantitative separation between these two Ansätze.

## 1 INTRODUCTION

Neural networks have proven very successful in parametrizing non-linear approximation spaces in high-dimensions, thanks to the ability of neural architectures to leverage the physical structure and symmetries of the problem at hand, while preserving universal approximation. The successes cover many areas of engineering and computational science, from computer vision (Krizhevsky et al., 2017) to protein folding (Jumper et al., 2021).

In each case, modifying the architecture (e.g. by adding layers, adjusting the activation function, etc.) has intricate effects in the approximation, statistical and optimization errors. An important aspect in this puzzle is to first understand the approximation abilities of a certain neural architecture against a class of target functions having certain assumed symmetry (LeCun et al., 1995; Cohen et al., 2018). For instance, symmetric functions that are permutation-invariant, ie $f(x_{\sigma(1)}, \ldots, x_{\sigma(N)}) = f(x_1, \ldots x_N)$ for all $x_1, \ldots, x_N$ and all permutations $\sigma : \{1, N\} \to \{1, N\}$ can be universally approximated by several neural architectures, e.g DeepSets (Zaheer et al., 2017) or Set Transformers (Lee et al., 2019); their approximation properties (Zweig & Bruna, 2022) thus offer a first glimpse on their efficiency across different learning tasks.

In this work, we focus on quantum chemistry applications, namely characterizing ground states of many-body quantum systems. These are driven by the fundamental Schröndinger equation, an eigenvalue problem of the form

$$H\Psi = \lambda\Psi \,,$$

where $H$ is the Hamiltonian associated to a particle system defined over a product space $\Omega^{\otimes N}$, and $\Psi$ is the *wavefunction*, a complex-valued function $\Psi : \Omega^{\otimes N} \to \mathbb{C}$ whose squared modulus $|\Psi(x_1, \ldots, x_N)|^2$ describes the probability of encountering the system in the state $(x_1, \ldots, x_N) \in \Omega^{\otimes N}$. A particularly important object is to compute the *ground state*, associated with the smallest eigenvalue of $H$. On Fermionic systems, the wavefunction satisfies an additional property, derived from Pauli's exclusion principle: the wavefunction is *antisymmetric*, meaning that

$$\Psi(x_{\sigma(1)}, \ldots, x_{\sigma(N)}) = \text{sign}(\sigma)\Psi(x_1, \ldots, x_N) \,. \tag{1}$$

The antisymmetric constraint is an uncommon one, and therefore demands unique architectures to enforce it. The quintessential antisymmetric function is a *Slater determinant* (Szabo & Ostlund, 2012), that we now briefly describe. Given functions $f_1, \ldots, f_N : \Omega \to \mathbb{C}$, they define a rank-one tensor mapping $f_1 \otimes \cdots \otimes f_N : \Omega^{\otimes N} \to \mathbb{C}$ by $(f_1 \otimes \cdots \otimes f_N)(x_1, \ldots, x_N) := \prod_{j \le N} f_j(x_j)$. The Slater determinant is then the orthogonal projection of a tensor rank one function into antisymmetric space. In other words, the rank one tensor $f_1 \otimes \cdots \otimes f_N$ is projected to $\mathcal{A}(f_1 \otimes \cdots \otimes f_N) := \frac{1}{N!} \sum_{\sigma \in S_N} (-1)^\sigma f_{\sigma(1)} \otimes \cdots \otimes f_{\sigma(N)}$. In coordinates, this expression becomes

$$\mathcal{A}(f_1 \otimes \cdots \otimes f_N)(x_1, \ldots, x_N) = \frac{1}{N!} \det \begin{bmatrix} f_1(x_1) & \ldots & f_1(x_N) \\ f_2(x_1) & \ldots & f_2(x_N) \\ & \ldots & \\ f_N(x_1) & \ldots & f_N(x_N) \end{bmatrix},$$

which shows that is antisymmetric following the alternating property of the determinant.

The Slater Ansatz is then simply a linear combination of several Slater determinants, of the form $F(x) = \sum_{l \le L} \mathcal{A}(f_1^l \otimes \cdots \otimes f_N^l)$, similarly as a shallow (Euclidean) neural network formed as a linear combination of simple non-linear ridge functions. While this defines a universal approximation class for antisymmetric functions (as a direct consequence of Weierstrass universal approximation theorems for polynomials), the approximation rates will generally be cursed by the dimensionality of the input space, as is also the case when studying lower bounds for standard shallow neural networks Maiorov & Meir (1998).

In the case of particles in $\Omega = \mathbb{R}$ or $\mathbb{C}$, it is classical that all antisymmetric functions can be written as a product of a symmetric function with the Vandermonde (see Section 3). This setting is generally considered much easier than settings with higher-dimensional particles, as this Vandermonde factorization no longer applies, though there are still ansätze that mimic this formulation (Han et al., 2019b).

A more powerful variant is the Jastrow Ansatz, where each Slater determinant is 'augmented' with a symmetric prefactor (Jastrow, 1955), ie $G = p \cdot \mathcal{A}(f_1 \otimes \cdots \otimes f_N)$ where $p$ is permutation-invariant. Clearly, $G$ is still antisymmetric, since the product of an antisymmetric function with a symmetric one is again antisymmetric, but grants more representational power. Other parametrisations building from Jastrow are popularly used in the literature, e.g. *backflow* (Feynman & Cohen, 1956), which models particle interactions by composing the Slater determinant with a permutation equivariant change of variables. Among practitioners, it is common knowledge that the Slater Ansatz is inefficient, compared to Jastrow or other more advanced parameterizations. Yet, there is no proven separation evinced by a particular hard antisymmetric function. We note that the Jastrow ansatz is strictly generalized by backflow (see Section 3), so separations between Slater and Jastrow would have immediate consequences for separations from the stronger architectures as well.

In this work, we are interested in understanding quantitative differences in approximation power between these two classes. Specifically, we wish to find antisymmetric target functions $G$ such that $G$ can be efficiently approximated with the Jastrow ansatz, i.e. approximated to $\epsilon$ error in the infinity norm with some modest dependence on the parameters $N$ and $\epsilon$ by a single Slater determinant with a single symmetric prefactor, yet no Slater representation can approximate $G$ for reasonably small widths. This question mirrors the issue of depth separation in deep learning theory, where one seeks functions that exhibit a separation between, for example, two layer and three layer networks (Eldan & Shamir, 2016), as well as recent separations between classes of symmetric representations (Zweig & Bruna, 2022).

**Main Contribution:** We prove the first explicit separation between the two ansätze, and construct an antisymmetric function $G$ such that:

- In some norm, $G$ cannot be approximated better than constant error by the Slater ansatz, unless there are $O(e^{N^2})$ many Slater Determinants.
- $G$ can be written in the Jastrow ansatz with neural network widths bounded by $poly(N)$ for specific activations, or in $N^{O(N)}$ using complex ReLU

## 2 RELATED WORK

### 2.1 MACHINE LEARNING FOR QUANTUM CHEMISTRY

Numerous works explore how to use neural network parameterizations effectively to solve Schrödinger's equation. These include works in first quantization, which try to parameterize the wavefunction $\Psi$ directly (Pfau et al., 2020; Hermann et al., 2020), and second quantization, where the wavefunction is restricted to an exponentially large but finite-dimensional Hilbert space, and then the problem is mapped to a spin system (Carleo & Troyer, 2017).

### 2.2 ANTISYMMETRIC ANSÄTZE

Numerous architectures enforce antisymmetry. In this work we focus primarily on the Slater ansatz and Jastrow ansatz, but others exist and are used in practice, with associated guarantees of universality (Han et al., 2019a). The backflow ansatz enables interaction between particles while preserving antisymmetry (Luo & Clark, 2019). More recently, an ansatz that introduces hidden additional fermions was introduced in Moreno et al. (2021).

### 2.3 ARCHITECTURE SEPARATIONS

A large body of work studies the difference in expressive power between different neural network architectures. These works frequently center on the representational gap between two-layer and three-layer networks (Eldan & Shamir, 2016; Daniely, 2017). Relatedly, several works have considered the representational power of different networks architectures constrained to be symmetric (Wagstaff et al., 2019; 2022; Zweig & Bruna, 2022).

The most closely related work to ours is Huang et al. (2021), which proves a non-constructive limit on the representability of the backflow ansatz, but requires exact representation rather than approximation in some norm. Conversely, Hutter (2020) demonstrates the universality of a single backflow ansatz, but requires a highly discontinuous term that may not be efficiently representable with a neural network.

## 3 PRELIMINARIES AND MAIN THEOREM

### 3.1 ANTISYMMETRIC ANSÄTZE

We consider $N$ particles restricted to the complex unit circle. That is, $x \in \Omega^N$ with $\Omega = \{z \in \mathbb{C}; |z| = 1\}$. We denote the tensor product $\otimes$ where, for $f, g : \Omega \to \mathbb{C}$, we have $f \otimes g : \Omega^2 \to \mathbb{C}$ such that $(f \otimes g)(x, y) = f(x)g(y)$. Given a permutation $\sigma \in \mathcal{S}_N$, and $x \in \Omega^N$, we denote by $\sigma.x = (x_{\sigma(1)}, \ldots, x_{\sigma(N)}) \in \Omega^N$ the natural group action.

Let $\mathcal{A}$ denote the antisymmetric projection, defined via:

$$\mathcal{A}(\phi_1 \otimes \cdots \otimes \phi_N) = \frac{1}{N!} \sum_{\sigma \in S_N} (-1)^\sigma \phi_{\sigma(1)} \otimes \cdots \otimes \phi_{\sigma(N)} \tag{2}$$

So up to rescaling we can consider Slater determinants as terms of the form $\mathcal{A}(\phi_1 \otimes \cdots \otimes \phi_N)$. Each $\phi_n$ is called an *orbital*. Intuitively, a Slater determinant is the simplest way to write an antisymmetric function, inheriting the antisymmetry property from the determinant itself.

Thus the Slater determinant ansatz with $L$ terms can be written as:

$$F = \sum_{l=1}^{L} \mathcal{A}(f_1^l \otimes \cdots \otimes f_N^l) . \tag{3}$$

Similarly, the Jastrow ansatz (with only one term) (Jastrow, 1955) can be written as:

$$G = p \cdot \mathcal{A}(g_1 \otimes \cdots \otimes g_N) \tag{4}$$

where $p$ is a symmetric function, namely $p(\sigma.x) = p(x)$ for any $\sigma$ and $x$. It is immediate to verify that $G$ is antisymmetric. Finally, the Backflow ansatz (Feynman & Cohen, 1956) (considering again a single term) is defined as

$$\widetilde{G}(x) = \mathcal{A}(\tilde{g}_1 \otimes \ldots \tilde{g}_N)(\Phi(x)), \tag{5}$$

where $\Phi : \Omega^N \to \widetilde{\Omega}^N$ is an equivariant flow, satisfying $\Phi(\sigma.x) = \sigma.\Phi(x)$, and where in general $\widetilde{\Omega}$ might be higher-dimensional than $\Omega$.

In particular, we verify that

$$\Phi : \Omega^N \to (\mathbb{C} \times \Omega)^N, \tag{6}$$
$$x \mapsto \Phi(x) := ((p^{1/N}(x); x_1), \ldots, (p^{1/N}(x); x_N)) \tag{7}$$

is equivariant. Given a collection of $N$ orbitals $\phi_1, \ldots \phi_N : \Omega \to \mathbb{C}$, we verify that the Jastrow Ansatz $G = p \cdot \mathcal{A}(g_1 \otimes \cdots \otimes g_N)$ can be written as $G = \mathcal{A}(\tilde{g}_1 \otimes \ldots \tilde{g}_N) \circ \Phi$, with $\tilde{g}_j : \mathbb{C} \times \Omega \to \mathbb{C}$ defined as $\tilde{g}_j(z, x) = z \cdot g_j(x)$. Thus, the Jastrow ansatz can be recovered as a particular case of the more general backflow ansatz. Therefore, quantitative separations between Slater and Jastrow automatically imply the same rates for Backflow.

## 3.2 INNER PRODUCTS

To measure the distance between the Slater Determinant ansatz and the Jastrow ansatz, we need an appropriate norm.

For univariate functions $f, g : S^1 \to \mathbb{C}$, define the inner product:

$$\langle f, g \rangle := \frac{1}{(2\pi)} \int_0^{2\pi} f(e^{i\theta})\overline{g(e^{i\theta})}d\theta . \tag{8}$$

For functions acting on $N$ particles, $F, G : (S^1)^N \to \mathbb{C}$, the associated inner product is

$$\langle F, G \rangle := \frac{1}{(2\pi)^N} \int_{[0,2\pi]^N} F(e^{i\boldsymbol{\theta}})\overline{G(e^{i\boldsymbol{\theta}})}d\boldsymbol{\theta} . \tag{9}$$

Let us introduce the notation that for $x \in \mathbb{C}^N$ and $\alpha \in \mathbb{N}^N$, $x^\alpha = \prod_{i=1}^N x_i^{\alpha_i}$. Then the orthogonality of the Fourier basis may be written as $\langle x^\alpha, x^\beta \rangle = \delta_{\alpha\beta}$.

## 3.3 THEOREM STATEMENT

With this, we may state our main result explicitly:

**Theorem 3.1.** *Consider a Slater ansatz with L terms:*

$$F = \sum_{l=1}^L \mathcal{A}(f_1^l \otimes \cdots \otimes f_N^l) \tag{10}$$

*parameterized by orbitals $f_n^l : S^1 \to \mathbb{C}$, and a Jastrow ansatz*

$$\hat{G} = p \cdot \mathcal{A}(g_1 \otimes \cdots \otimes g_N) \tag{11}$$

*parameterized by orbitals $g_n : S^1 \to \mathbb{C}$ and a symmetric Jastrow factor $p : (S^1)^N \to \mathbb{C}$.*

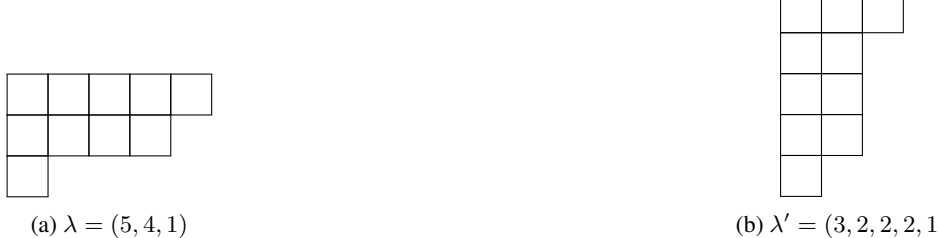

(a) $\lambda = (5, 4, 1)$  (b) $\lambda' = (3, 2, 2, 2, 1)$

Figure 1: Example of Young diagram and conjugate partition

*Let $N \geq 3$ and $1 > \epsilon > 0$. Then there is a hard function $G$ with $\|G\| = 1$, such that $\hat{G}$ parameterized by neural networks with width, depth, and weights scaling in $O(poly(N^N, \epsilon))$ that can approximate $G$:*

$$\|\hat{G} - G\|_\infty < \epsilon \tag{12}$$

*but, for a number of Slater determinants $L \leq e^{N^2}$:*

$$\min_F \|F - G\|^2 \geq \frac{3}{10} \;. \tag{13}$$

Note the constraint $\|G\| = 1$ is just to prevent vacuous solutions by scaling $G$ to be arbitrarily big. We will describe the exact network structure of $G$ in Section A.4.

We also remark that $N^{O(N)} \ll e^{N^2}$, so this is a true separation. Furthermore, the construction only requires $N^{O(N)}$ parameters due to the analysis of a generic complex ReLU activation. For particular choices of the activation, the network only requires $poly(N)$ parameters. Finally, the restriction of $\hat{G}$ to only have one determinant is artificial, here to demonstrate the nature of the separation. In practice, the Jastrow ansatz allows for multiple determinant terms in learning.

## 3.4 SCHUR POLYNOMIALS

To build up the difficult function $G$, we use several identities related to the symmetric Schur polynomials. First, we introduce partitions as they will be used to index Schur polynomials:

**Definition 3.2.** *An* integer partition $\lambda$ *is non-increasing, finite sequence of positive integers $\lambda_1 \geq \lambda_2 \geq \cdots \geq \lambda_k$. The weight of the partition is given by $|\lambda| = \sum_{i=1}^k \lambda_i$. The length of a partition $l(\lambda)$ is the number of terms in the sequence. We call a partition* even *if every $\lambda_i$ is even.*

Partitions can be represented by their Young diagram, see Figure 1. Furthermore, we will need the notion of a conjugate partition:

**Definition 3.3.** *Given a partition $\lambda$, the* conjugate partition $\lambda'$ *is gotten by reflecting the Young diagram of $\lambda$ along the line $y = -x$. We call a partition* doubly even *if $\lambda$ and $\lambda'$ are both even.*

First, we introduce the Vandermonde written as:

$$V(x) = \prod_{i<j} (x_j - x_i) \;. \tag{14}$$

Then we denote the Schur polynomial indexed by partition $\lambda$ as:

$$s_\lambda(x) := \begin{cases} V(x)^{-1} \det \left[ x_i^{\lambda_j + N - j} \right] & l(\lambda) \leq N \;, \\ 0 & l(\lambda) > N \;. \end{cases} \tag{15}$$

Given two partitions $\lambda$ and $\mu$, the following fact follows easily from linearity of the determinant:

$$\langle s_\lambda \cdot V, s_\mu \cdot V \rangle = N! \cdot \delta_{\lambda\mu} \;. \tag{16}$$

We will in the sequel assume $N$ is even, then we can cite the following formal identity of a particular Pfaffian:

**Theorem 3.4** (Sundquist (1996) Theorem 5.2, Ishikawa et al. (2006) Corollary 4.2).

$$\sum_{\lambda \; doubly \; even} s_\lambda \cdot V = \text{Pf} \left[ \frac{x_i - x_j}{1 - x_i^2 x_j^2} \right] \tag{17}$$

$$= \prod_{i<j} \frac{1}{1 - x_i^2 x_j^2} \cdot N! \cdot \mathcal{A}(\phi_1 \otimes \cdots \otimes \phi_N) , \tag{18}$$

*where we set the $\phi$ maps to be:*

$$\phi_j(x_i) = \begin{cases} x_i(x_i^2)^{N/2-j}(1 + x_i^4)^{j-1} & 1 \le j \le N/2 \\ (x_i^2)^{N-j}(1 + x_i^4)^{j-1-N/2} & N/2 + 1 \le j \le N \end{cases} \tag{19}$$

## 4 PROOF SKETCH

We give here a sketch of the tactic of the proof. If we simplify the problem to a question of antisymmetric tensor products of vectors instead of functions, then inapproximability of Slater determinants amounts to finding a high-rank antisymmetric tensor, that cannot be approximated in the $l_2$ norm by a small number of rank-one tensors.

The usual trick for this problem is to flatten all the tensors and rewrite them as matrices, so the problem reduces to approximating a high rank matrix by a low rank one, which is solved by SVD. To make the SVD tractable, the high-rank matrix is typically taken to be diagonal.

In our setting, we cannot simply choose a tensor that will be diagonal after flattening, because the constraints of antisymmetry will enforce certain matrix elements to be equal. It turns out we can focus on a particular subtensor, where it's possible to flatten to a diagonal matrix, while nevertheless keeping our hard function representable by the Jastrow ansatz.

Indeed, the hard function $G$ can be written exactly in the Jastrow ansatz by the above identity:

$$G := \frac{C}{\sqrt{N!}} \sum_{\lambda \; doubly \; even} r^{\left(|\lambda| + \frac{N(N-1)}{2}\right)} s_\lambda \cdot V = C\sqrt{N!} \cdot \prod_{i<j} \frac{1}{1 - r^4 x_i^2 x_j^2} \cdot \mathcal{A}(\phi_1^{(r)} \otimes \cdots \otimes \phi_N^{(r)}) , \tag{20}$$

for some choice of $\phi_j$ maps and $C, r$ with $|r| < 1$.

Let us explain the signficance of restricting the support to doubly even Schur functions. Let $\delta = (N - 1, N - 2, \ldots, 1, 0)$. Then by simply canceling the Vandermonde factor, for an appropriate partition $\lambda$, we have:

$$s_\lambda(x) \cdot V(x) = \det \left[ x_i^{\lambda_j + \delta_j} \right]$$

Furthermore, if $\lambda$ is doubly even, then $\lambda + \delta$ will take the form $(2a + 1, 2a, 2b + 1, 2b, \ldots)$ with alternating odd and even terms with the odd term one above the subsequent even term. See Figure 2 for an example.

The significance of this structure is that, by knowing only the odd values of $\lambda + \delta$, the even values are determined. We will make essential use of this property to flatten an antisymmetric tensor to a matrix with sets of odd indices as rows, and sets of even indices as columns. For this matrix, the above structure implies diagonality, and from there we can proceed with a usual proof of low-rank approximation from SVD.

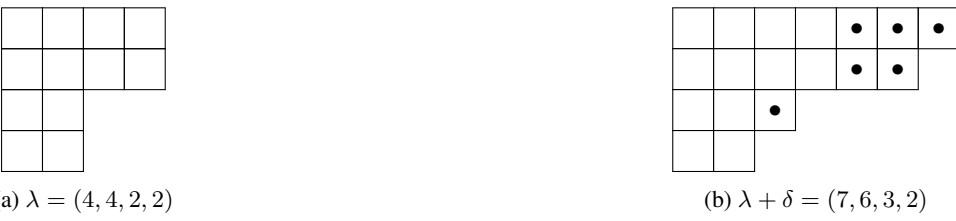

(a) $\lambda = (4, 4, 2, 2)$        (b) $\lambda + \delta = (7, 6, 3, 2)$

Figure 2: $\lambda$ and $\lambda + \delta$ for $\lambda$ doubly even.
Note that $\lambda + \delta \sim (6 + 1, 2 + 1) \cup (6, 2)$.

## 5 EXPERIMENTS

We illustrate the nature of this exponential separation in finite case, specifically by seeking to learn our hard function in the Slater ansatz and Jastrow ansatz with $N = 4, 6$ particles. In particular, we try to learn the hard function $G$ with the two ansätze (rescaled with the $C$ term as this constant is extremely small to give normalization in the $L_2$ norm).

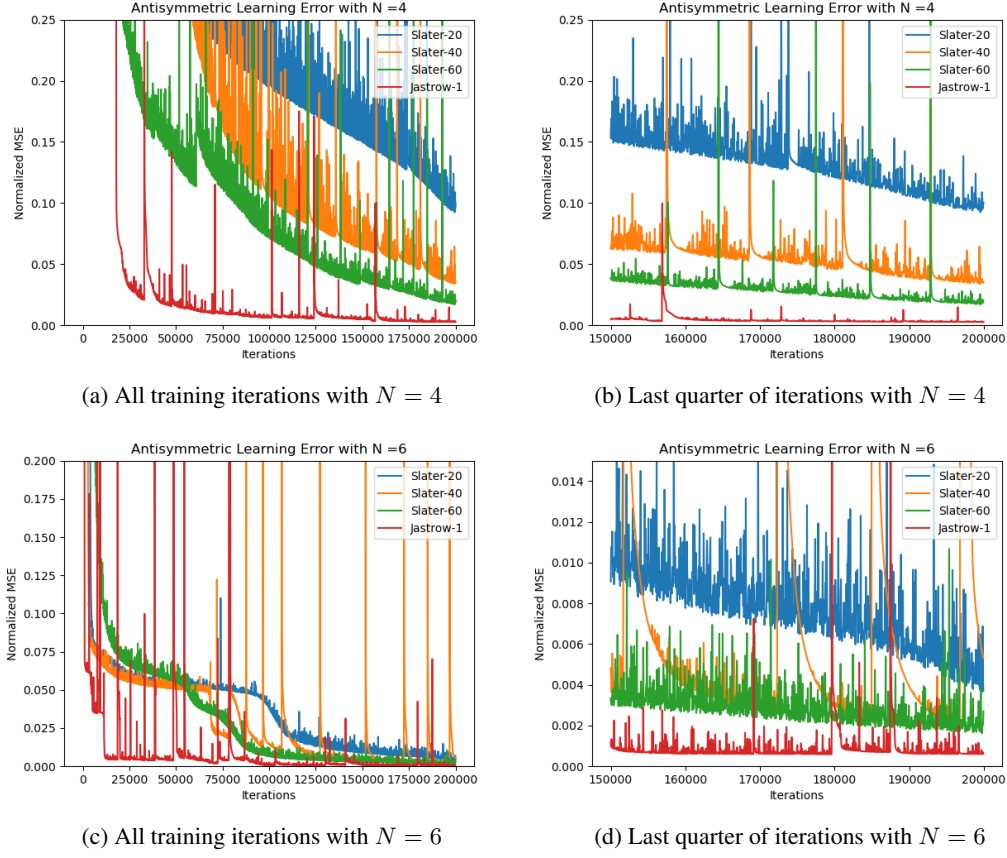

(a) All training iterations with $N = 4$      (b) Last quarter of iterations with $N = 4$

(c) All training iterations with $N = 6$      (d) Last quarter of iterations with $N = 6$

Figure 3: Training MSE for Slater ansatz of varying number of determinants and Jastrow ansatz of one determinant.

We consider the empirical approximation of mean squared training error in $\| \cdot \|$, where we compare learning with the Slater ansatz and a large number of determinants, vs. the Jastrow ansatz with a single determinant. We parameterize each orbital with a three layer neural network with hidden width 30, and activation the complex ReLU that maps $a + bi \mapsto ReLU(a) + ReLU(b)i$. We parameterize the Jastrow term with the Relational Network (Santoro et al., 2017) and multiplication

pooling where all networks are three layers and all hidden widths are also 30. Learning rate is set to 0.0005 in all runs, for 200000 iterations of full batch gradient descent on 10000 samples drawn i.i.d. from the complex unit circle. We plot the normalized MSE, i.e. MSE divided by the error attained by the naive zero function.

The results are given in Figure 3. Each Slater ansatz is labeled by the number of determinants it is parameterized with. We observe that one Jastrow determinant suffices where a large number of Slater determinants fails to achieve low MSE, alluding to the exponential nature of the separation as the number of particles $N$ increases.

# 6 DISCUSSION

## 6.1 PROOF LIMITATIONS

The proof technique relies on finding a symmetric function that is supported exclusively on doubly even Schur polynomials. This is established in the Pfaffian identity given in Theorem 3.4. This yields a function that requires exponentially many Slater determinants but may be written exactly in Jastrow form.

However, the large magnitude of the Jastrow factor precludes efficient approximation in the infinity norm. This cannot be overcome by changing the value of $r$: as $r$ approaches 1, the magnitude of support on high dimensional doubly even Schur polynomials increases while simultaneously the magnitude of the Jastrow factor increases. So we must choose a sufficiently large $r$ in order to guarantee the induced matrices are effectively high-rank.

An alternative tactic would be to control approximation in the $L_2$ norm given by $\| \cdot \|$. However, calculating the $L_2$ norm is most easily done after decomposing into the orthogonal basis of multinomials, which is challenging when multiplying terms together in the Jastrow product.

This proof also only uses 1-dimensional particles to evince a separation. A nearly identical proof could be employed for higher-dimensional particles by only utilizing the first component, but there would be no dependence on the dimension $d$. Understanding the simultaneous dependence on $N$ and $d$ would therefore require a new proof technique.

## 6.2 OPEN QUESTIONS

The main result of this work represents a first step in understanding separations between relevant antisymmetric ansätze. We conclude with a discussion of the natural open questions in this domain.

**Stronger Separation**    It would be ideal to strengthen this bound to show polynomial efficiency in approximating with the Jastrow ansatz, or to alternatively prove this is not possible.

**Practical Wavefunctions**    Extending the analysis to consider wavefunctions that appear in more practical applications would be informative. For example, in the one-dimensional case, famously the eigenfunctions of the Sutherland model are known (Langmann, 2005), and the representability of these functions in particular is an open question.

**More Powerful Ansätze**    The separation we demonstrate in this work is between the two simplest ansätze. Demonstrating separations among the more expressive network architectures, for example the backflow or hidden fermion models discussed previously, would prove the merit of these more complicated and time-intensive methods.

**Learnability separations** Our current separation concerns exclusively the approximation properties of the two parametric families of antisymmetric functions, and as such neglects any optimization question. It would be interesting to integrate the optimization aspect in the separation, similarly as in (Safran & Lee, 2022) for fully-connected networks.

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
