# OpenReview forum: "Towards Antisymmetric Neural Ansatz Separation"
_ICLR.cc/2023/Conference — Submitted to ICLR 2023_

### Official Review · Reviewer_Gasa · 2022-10-21

**Confidence:** 4
**Correctness:** 4
**Technical Novelty And Significance:** 3
**Empirical Novelty And Significance:** 2
**Recommendation:** 5

**Clarity, Quality, Novelty And Reproducibility:**

The paper is written clearly. The main theoretical contribution is indeed
novel, and the experiments appear to be reproducible.

**Strength And Weaknesses:**

**Strengths:**
- This work constructs an antisymmetric function that provably shows a
  difference in the expressiveness of the Slater and Jastrow ansatze given a
  comparable number of variables.
- The techniques used for this result use clean and powerful tools from
  algebraic combinatorics.

**Weaknesses:**
- This is primarily a math/mathematical physics paper. There are connections to
  representation learning (i.e., representing the target function via neural
  network-based ansatze), but they seem limited in scope.

**Suggestions:**
- [page 2] suggestion: It might be slightly more direct to first give the
  mathematical definition of *Slater determinant*, and then describe what is
  happening in words. The determinant-based definition is very direct and easy
  to parse.
- [page 7] Given the nature of this paper, the experiments used to explore
  Theorem 3.1 are very under-developed. The main results are interesting and
  exciting, but this does not fully come through from the experiments.

**Summary Of The Paper:**

This work studies a quantitative separation between two fundamental models of
anti-symmetric functions in quantum chemistry: The Slater ansatz and Jastrow
ansatz. The authors construct a target function (Equation 20) using a subtle
property of Schur polynomials to show that the Jastrow ansatz can approximate
this function with exponentially fewer variables than the Slater ansatz. This
gives the first explicit separation between the two ansatze. Finally, the
authors begin to illustrate this separation property empirically by fitting a
modified version of this target function with neural network-based relaxations
of the two ansatze.

**Summary Of The Review:**

This paper makes a nice theoretical contribution about the difference in the
representation power of the Slater ansatz and Jastrow ansatz. A witness function
is constructed using advanced techniques from algebraic combinatorics, and some
experiments are included to illustrate this difference. Overall, this is a good
paper but it does not seem appropriate for ICLR as written.

---

> ### Author Response · Authors · 2022-11-17
> **Rebuttal**
>
> Thank you for your review, we address questions and concerns below.
>
> Thanks for the suggestion, we agree the Slater ansatz could use a bit more exposition when we introduce it.
>
> We also feel the experiments as given before were somewhat unsatisfying.  As discussed in the response to reviewer PU8s, we have updated our experimental setup to now use the true hard function (up to a choice of scaling), which we feel should make the empirical results a more honest reflection of the theory.
>
> On the point of the theoretical contribution and its appropriateness for ICLR, we would argue that the ICLR call for papers includes "theoretical issues in deep learning".  As discussed in the response for reviewer Cp2s, there are many new ansätze applied in ML for quantum chemistry, and this work represents a first step in understanding the expressive power of these models.

---

> > ### Comment · Reviewer_Gasa · 2022-11-28
> > **Followup response**
> >
> > Thank you to the authors for all of your reviewer responses and for updating the experiments. I increased my score from 3 --> 5.

---

### Official Review · Reviewer_Cp2s · 2022-10-22

**Confidence:** 2
**Clarity, Quality, Novelty And Reproducibility:** The paper is clear and the contributi…
**Correctness:** 4
**Technical Novelty And Significance:** 2
**Empirical Novelty And Significance:** Not applicable
**Recommendation:** 6

**Strength And Weaknesses:**

The main result of this paper is a “hard” antisymmetric function that has the following properties:

(a) the function cannot be approximated better than some constant with $o(\exp(N^2))$ many Slater determinants, where $N$ is the number of particles (i.e., dimension of the space). This means that its Slater ansatz is quite inefficient.

(b) the function can be written in the Jastrow ansatz using a single Slater determinant and a prefactor which is parametrized by neural networks of width, depth and weights of order either $O(N^N)$ with ReLU activations or $\mathrm{poly}(N)$ with other activations.

The paper is well-written and its contribution is clear. The hard instance's construction seems nice and relies on some known tools from representation theory. Essentially, the provided separation adds to other results related to architecture separations in terms of approximation power. Regarding technical novelty, the only contribution lies in the construction of the hard instance. However, it is not clear to me whether the construction and the technical novelty can place the paper higher than marginally above the acceptance threshold. It would be nice if the authors could further motivate the studied problem.

In short, I believe that the paper is marginally above the acceptance threshold, yet I am willing to increase my score depending on authors' response.

I would like to close with some questions:

Q1. Could the authors provide some ML applications of these antisymmetric ansatze? Do they only arise in quantum chemistry applications?

Q2. Could the authors further comment on the last open question they propose?

**Summary Of The Paper:**

This paper studies quantitative separations between two families of antisymmetric functions: The Slater Ansatz which is a linear combination of Slater determinants and the more powerful Jastrow Ansatz where each Slater determinant is augmented by a symmetric prefactor (and hence remains antisymmetric). In this area of research, it is widely known that Jastrow ansatz is more efficient (in terms of expressivity) compared to Slater, yet no separation was known. The paper provides such a separation in terms of expressivity power.


**Summary Of The Review:**

The paper deals with the approximation power of some families of antisymmetric functions and provides a separation between two well-known families. I believe that the contribution is marginally beyond acceptance, since the separation by itself and the technical work in order to obtain it do not seem sufficient to give a higher score to the paper.

---

> ### Author Response · Authors · 2022-11-17
> **Rebuttal**
>
> Thank you for your review, we address your questions and comments below.
>
> As to motivation, we are driven by the broad spectrum of new antisymmetric ansätze in the literature, including backflow-driven models like FermiNet[1] and PauliNet[2], the hidden fermion model[3], models based on Pfaffians[4], etc.  These models differ on several axes, in terms of time complexity, empirical expressiveness, and qualitative performance on different molecules.
>
> Very little is known about provably differences in expressiveness between these models; we consider this work a first step towards understanding comparison of these models, beginning with Slater vs Jastrow as even this simpler case was previously unstudied.
>
> To answer Q1: yes, the primary application of antisymmetric ansätze is in fermionic systems in quantum chemistry, of which the four papers cited are several examples of applying these models in ML for chemistry.
>
> To answer Q2: our last open question (about learnability of our hard function) is difficult because even the broader question of learning any non-trivial antisymmetric function is unknown.  In particular, say learning the Vandermonde is not obvious.  Though there are results about learnability of some functions with 2-layer neural networks, a Slater determinant where each orbital is a 2 layer network is far more complicated structurally than a 2 layer network itself, so it's not yet clear what results can be transported to the antisymmetric setting.
>
>
>
> [1] Spencer, James, et al. "Ab-Initio Solution of the Many-Electron Schrödinger Equation with Deep Neural Networks." Bulletin of the American Physical Society 65 (2020).
>
> [2] Hermann, Jan, Zeno Schätzle, and Frank Noé. "Deep-neural-network solution of the electronic Schrödinger equation." Nature Chemistry 12.10 (2020): 891-897.
>
> [3] Robledo Moreno, Javier, et al. "Fermionic wave functions from neural-network constrained hidden states." Proceedings of the National Academy of Sciences 119.32 (2022): e2122059119.
>
> [4] Bajdich, M., et al. "Pfaffian pairing and backflow wavefunctions for electronic structure quantum Monte Carlo methods." Physical Review B 77.11 (2008): 115112.

---

> > ### Comment · Reviewer_Cp2s · 2022-11-24
> > **Thanks for the response.**
> >
> > First, I would like to thank the authors for the response. As far as my score is concerned, I would like to keep it unchanged.

---

### Official Review · Reviewer_PU8s · 2022-10-25

**Confidence:** 2
**Correctness:** 2
**Technical Novelty And Significance:** 4
**Empirical Novelty And Significance:** 2
**Recommendation:** 6

**Clarity, Quality, Novelty And Reproducibility:**

The writing is mostly clear.
The topic and theory appear novel to me.

**Strength And Weaknesses:**

The topic of this paper is interesting and the writing is mostly clear.

However I have concerns about the soundness of theory and experiments.

1.
After Theorem 3.1, authors claim that "For particular choices of the activation, the network only requires $poly(N)$ parameters." However, I cannot find a proof in the main paper and supplement.

2.
In the experiment, authors don't use the hard function they choose in theory because "the Jastrow term $
\prod_{i<j} \frac{1}{1-r^4 x_i^2 x_j^2}
$ can lead to very large function values". On the other hand, their theory guarantee that $\lVert G\rVert=1$, which indicates that function values should not be too large. Why there is such seemingly disagreement?

3.
In the experiment, two figures for $N=4$ and $N=6$ are shown. In the supplementary material, another picture called `plot_2.png` is also shown but not included in the main paper. This experiment shows small gap between Slater ansätze and Jastrow ansätze. Why there is only large gap in $N=4$ experiment?

Typos:
page 6: some choice of -> same choice of.
page 6: sign[i]ficance.
page 7: hidden weights -> hidden width.


**Summary Of The Paper:**

This paper studies the separation of two antisymmetric ansätze, i.e. Slater ansätze and Jastrow ansätze.

**Summary Of The Review:**

Separation of antisymmetric ansätze is an interesting topic. However, several claims in this paper is not fully justified.

---

> ### Author Response · Authors · 2022-11-17
> **Rebuttal**
>
> Thank you for your review, we address your concerns below.
>
> You are correct that we mentioned the proof of Theorem 3.1 using specific activations but were not explicit in proving it.  We have rectified this with a more explicit discussion of this fact (essentially the proof amounts to taking the Jastrow terms and orbitals in the defintion of the hard function G and considering the somewhat artificial setting where the activations are exactly chosen equal to these functions).
>
> We agree that using a surrogate hard function is somewhat unsatisfying in our experiments, and we have updated all experiments to use the true hard function (up to a choice of scaling), which we feel should make the empirical results a more honest reflection of the theory.
>
> To your point about the norm of the Jastrow factor, we note that our Jastrow factor is normalized to have L_2 norm equal to one, but this does not automatically control the infinity norm, which indeed can reach exponentially large values.
>
> The plot_2.png in the supplement was an unused result under the choice of N = 2 set elements.  We omitted this plot because as you are correct that at such a small set size the separation result is not clear: because the separation is exponential in N, this fact is not surprising and the plots for larger values of N are more important for evincing the separation.
>
> Thank you for pointing out the typos!

---

> > ### Comment · Reviewer_PU8s · 2022-11-18
> > **I Appreciate Authors' Response**
> >
> > Their comment resolves my concern therefore I change my score to 6.

---

### Decision · Program_Chairs · 2023-01-20

**Decision:**

Reject

**Justification For Why Not Higher Score:**

Please see the above.


**Justification For Why Not Lower Score:**

N/A


**Metareview: Summary, Strengths And Weaknesses:**

Previous authors have developed parameterized function models specialized to capture structure in estimation problems arising in quantum chemistry.  Two prominent examples of these, which can be viewed as neural network models, are called the Slater Ansatz and Jastrow Ansatz.  This paper provides a separation result on the expressive power of these models, describing a function that can be approximated much more efficiently using the Jastrow Ansatz than with the Slater Ansatz.  They illustrate this separation through simulation experiments.

The fundamental novelty of the analysis was assessed to be limited.  The authors did not do enough to motivate this research for an ICLR audience.